# The Potential of Exploiting Economical Solar Dryer in Food Preservation: Storability, Physicochemical Properties, and Antioxidant Capacity of Solar-Dried Tomato (*Solanum lycopersicum*) Fruits

**DOI:** 10.3390/foods10040734

**Published:** 2021-03-30

**Authors:** Salah A. Al Maiman, Nawal A. Albadr, Ibrahim A. Almusallam, Mohammed Jawad Al-Saád, Sarah Alsuliam, Magdi A. Osman, Amro B. Hassan

**Affiliations:** 1Department of Food Science and Nutrition, College of Food and Agricultural Sciences, King Saud University, P.O. Box 2460, Riyadh 11451, Saudi Arabia; smaiman@ksu.edu.sa (S.A.A.M.); nalbader@ksu.edu.sa (N.A.A.); sarona-66@hotmail.com (S.A.); magdios@ksu.edu.sa (M.A.O.); 2Date and Palm Center, Ministry of Environment, Water and Agriculture, P.O. Box 43, Alhufuf 31982, Saudi Arabia; iam199@hotmail.com; 3Saudi Irrigation Organization, P.O. Box 43, Alhufuf 31982, Saudi Arabia; mohj1403@gmail.com

**Keywords:** solar drying, shelf life, bioactive compounds, tomato slices

## Abstract

This study investigated the effect of solar drying on storability and physiochemical and antioxidant capacities of dried tomatoes. Sliced fruit was dried at 45 ± 2 °C for 24 h under a solar tunnel dryer and stored at ambient temperature for 90 and 180 days. Solar drying treatments significantly (*p* < 0.05) reduced the bacterial and mold load, and eliminated *Staphylococcus aureus*, *S. saprophyticus*, and *Escherichia coli* in all samples. Solar drying treatment reduced the water activity of the dried tomato’s to 0.31 that remained at the same level during storage period 180 days. Storage of dried tomato slices resulted in the decrease of both color and vitamin C content while it increased the total carotenoid, lycopene, phenolic compound content, and antioxidant activity. Furthermore, the principle component analysis (PCA) revealed that solar drying of tomato slices enhanced its physicochemical properties, antioxidant capacity particularly after storage for 90 and 180 days. Interestingly, the solar drying process enhanced tomato slices storage and physicochemical characteristics, that resulted in extending the shelf life by up to 6 months, indicating the great potential application of low-tech solar in food industry and could become an emerging effective post-harvest preservative method for seasonal perishable vegetable and fruit, particularly in developing countries.

## 1. Introduction

Tomato (*Solanum lycopersicum*) and tomato-based products play an important role in human nutrition and health. Tomatoes are an excellent source of many nutrients and phytochemical compounds, such as minerals, vitamin C and E, phenolic flavonides, lycopene, and carotenoids [1,2]. Many studies had shown strong correlation between regular consumption of tomatoes and related products and prevention of chronic disease such as cancer and heart disease [3,4,5]. These therapeutic actions are mainly be due to the valuable bioactive components and antioxidant properties of the tomato slices [6,7].

Like other highly seasonal, perishable fruits, tomatoes are susceptible to microbial degradation and rottening that result in a loss of quality and shortening of the shelf life, leading to high post-harvest losses [8,9]. The post-harvest losses of seasonal fruits and vegetables were estimated to be between 40% and 50% in many developing countries [10]. Therefore, several post-harvest processing methods have been explored to maintain seasonal fruit quality, safety so as to extend their shelf life. Cold storage is commonly used to improve shelf life and maintain quality after harvest. However, in developing countries due to poor storage systems, poor transportation coupled with inadequate processing facilities, and lack of processing enterprises, considerable amounts of harvested tomatoes are lost [11]. Therefore, suitable preservation approaches with low-cost processing methods is required.

Drying is the earliest and most common technology used for food preservation. Drying is known to reduces the moisture content and water activity, thereby controlling microbial growth (bacteria, yeasts, and molds) and oxidative and enzymatic reactions to a minimal level, that results in safe storage and increase product shelf life [12,13,14,15]. Furthermore, decreasing the moisture content of the products reduces their weight, volume, facilitating packaging, storage, and transportion [16].

Among drying techniques, sun drying has the lowest processing costs and does not require expertise. However, open sun drying is prevalent for long drying duration, weather dependence, exposure to rain, dust, insects, and animals are major obstacles to sun drying together with color degradation, poor rehydration, and low quality of the end product. On the other hand, Solar dryers are evolved to save time and to maintain the quality of products. Moreover, solar dryers use a clean and non-conventional source of energy [17]. The use of a solar dryer can be ysed to enhance the safety and quality issues associated with sun-dried food products [18]. In addition, a solar dryer can rapidly dry the product consistently and hygienically to meet consumer demands [19]. Several studies have investigated the impact of the drying process, particularly solar drying, on the physical, chemical, and nutritional characteristics of fruits and vegetables. Abrol et al. [20], investigated the effect of solar tunnel drying on the physicochemical and antioxidant properties of tropical fruits mango, banana, and papaya and reported that solar tunnel drying increased physicochemical and antioxidant properties of the fruit while slightly decreasing the vitamin C content. Azeez et al. [21] reported that drying of tomato slices in a vaccum oven at different temperatures and times significantly increased antioxidant activity, phenolic compounds and lycopene but significantly drecreased flavonoids and β cartone content.

Despite several studies investigating the effect of solar energy drying on fruit quality, further research in this field is needed to improve the technology use in the food industry. To date, no study has been carried out to evaluate the changes in storability characteristics and physicochemical and antioxidant properties of solar-dried tomatoes. Therefore, this study is designed to assess the changes in microbial load, physico-chemical, antioxidant activity, and bioactive compounds change during the storage of solar-dried tomatoes.

## 2. Materials and Methods

### 2.1. Fruit Samples

Tomato samples were obtained from three tomato farm located in Khartoum, Sudan. After harvesting and sorting from the damaged ones, the fresh fruit samples were washed with tap water, and all inedible parts were removed manually.

### 2.2. Standards and Chemicals

Polyphenol standards (tannic acid, gallic acid, and catechin) with purity >97.5%, Folin–Ciocâlteu reagent, oxalic acid, aluminum chloride, sodium nitrite, ascorbic acid, 2,6-dichlorophenol indophenol dye, and 2,2-diphenyl-1-picrylhydrazyl (DPPH) were purchased from Sigma–Aldrich Company Ltd. (Darmstadt, Germany). All other chemicals used were of at least analytical grade.

### 2.3. Solar Drying

The prototype of tunnel dryer had dimension of 2 by 1 m and 1 by 1 m for collector and drying chamber respectively Both collector and the drying units were covered with UV stabilized visqueen sheets and food grade black paint was used as an absorber in the collectors. A photovoltaic solar panel (model BP 255) was used to power an axial fan of 12 V each to generate an air flow of 0.25 m^3^ s^−1^ through the drying bed. The temperature, humidity, and air velocity along the drying length and across the tunnel were measured under no load conditions. The temperature and humidity sensor locations in the dryer are given in five place along the dryer. Fruits were cut into angular slices with 3 mm in thickness. The slices were placed on a wire mesh tray in cabinet dryers in a single layer and directly kept under a solar tunnel dryer at a temperature of 45 ± 4 °C for 24 h. The dried tomatoes were backed into vacuum plastic bags and stored at room temperature 25 ± 2 °C for 90 and 180 days, respectively. Fresh fruit was analyzed as undried samples and served as the control.

### 2.4. Moisture Content and Water Activity

The moisture content of the fresh and solar-dried tomato was measured using the hot air method as previously described [22]. The water activity was determined in fresh and dried tomato slies was measured at 25 °C (±0.2 °C) using hygrometer with selectable sensors for determination of air humidity, material moisture, and water activity (humimeter RH2, Schaller, Vienna, Austria), equipped with a temperature-controlled system which allow to have a temperature stable sampling environment.

### 2.5. Microbial Load Determination

The microbial load of fresh and solar-dried tomato slices was enumerated by counting of *Staphylococcus aureus*, *S. saprophyticus, Escherichia coli*, yeasts, and molds (log cfu/g). Samples (1 g) were homogenized in 10 mL peptone water (0.1%) and serially diluted and plated using the appropriate medium. The total *Staphylococcus aureus* and *S. saprophyticus* count were plated in Petri films (Staph. express. IDEXX) and incubated at 37 ± 1 °C for 24 to 48 h as previously described [23]. *E. coli* was counted according to American Public Health Association (APHA) [24] using MacConkey agar medium. Yeasts and molds were counted on acidified potato dextrose agar, (OxoidCM139) which was acidified by the addition of the proper amount of sterile 10% tartaric acid (Fluka-AG-Buchs.SG). Solidified pour plates were incubated upright at 25 ± 1 °C for 3–7 days and results were reported as log per gram of samples according to Association of Official Analytical Chemists AOAC [22].

### 2.6. Color and Browning Index

The color of powdered samples was determined using CIELAB (CIE) color scales. *L** (whiteness/darkness), *a** (redness/greenness), and *b** (yellowness/blueness) values were measured using a colormeter (CR-300, Konica Minolta Inc., Tokyo, Japan). Five measurements were averaged, and the total color difference (Δ*E*) estimated using the following equation reported by [25].
1$$\Delta E=\sqrt{{\left(\Delta L\right)}^{2\;}+{\left(\Delta a\right)}^{2}+{\left(\Delta b\right)}^{2}}$$

Browning index (BI) was calculated from *L*, *a*, and *b* values as described by Maskan [26] following equation
2$$\mathrm{Browning}\;\mathrm{index}\;\left(\mathrm{BI}\right)=\frac{100\left(x-0.31\right)}{0.17},\;\mathrm{where}\;x\;=\frac{a+1.75L}{5.645L+\left(a-3012b\right)}$$

### 2.7. Titratable Acidity

Acidity was measured for the sample after mixing 1 g of sample with 10 mL of distilled water and then titrated against NaOH (0.1 N) until the color was pale pink. The titratable acidity was expressed as percent malic acid [22]:LA % = [10 × V. NaOH × 0.009 × 0.1/W] × 1003

### 2.8. Vitamin C Determination

Vitamin C in fresh and solar dried tomato slices was determined by titration with the indicator dye 2,6-dichloroindophenol according to the method of AOAC [22]. Approximately 5 g of sample was blended with 0.4% oxalic acid and filtered, and the volume adjusted to 100 mL with oxalic acid. A volume of 10 mL of this prepared sample was added to 5 mL of 10% oxalic acid and then titrated against 2,6-dichlorophenol indophenol dye.

### 2.9. Carotenoids Determination

Carotenoids were extracted according to the method of Jacques et al. [27]. Approximately 2 g tomato powder was weighed and homogenized with 25 mL of cold acetone. The mixture was shaken for 10 min at room temperature, followed by filtration using Whatman No. 1 filter paper. The supernatant was transferred to a decanting funnel, where a liquid:liquid extraction was performed with 20 mL petroleum ether. The filtrate was washed with 100 mL distilled water to remove the acetone, and the lower phase was discarded. The procedure was repeated twice, and the petroleum ether layer filtered using Whatman No. 1 filter paper covered with 5 g of anhydrous sodium sulfate to remove residual water. The petroleum ether extracts were obtained, and the volume was adjusted to 25 mL with petroleum ether. The absorbance was measured at 450 nm, and the total carotenoid content was expressed in mg/100 g dry matter DM.

### 2.10. Lycopene Determination

Lycopene in the tomato samples was extracted with a solvent mixture of hexane:ethanol:acetone (2:1:1, *v*/*v*) following the procedure of Sadler et al. [28] with minor modifications. Briefly, 1 g lyophilized sample was resuspended with 8 mL of the solvent mixture, capped and placed on the rotary mixer, then after at least 30 min of extraction, 10 mL of distilled water was added to separate the phases, and the absorbance of lycopene level was measured at 503 nm after 5 min. The lycopene content was expressed in mg/100 g DM.

### 2.11. Phenolic Content and In Vitro Antioxidant Activity

#### Extraction of Antioxidants

A methanolic extract of tomato samples was prepared at a ratio of 1:25 (*w*/*v*) at 25 °C overnight. The extract was filtered, and the process was repeated with the residue. Extracts were collected and dried using a vacuum with a rotary evaporator and then kept dry for further analysis. The extracts were reconstituted with pure ethanol directly before the analysis of total phenolic content, total flavonoids, and DPPH scavenging activity.

### 2.12. Determination of Total Phenolic Content

The Folin–Ciocâlteu method [29] was used to estimate the total phenolic content (TPC) of the fresh and solar dried tomato samples. An aliquot (20 µL) of the dried methanolic extract solution (1:10 *w*/*v*) was mixed with 1.58 mL H_2_O and 100 µL of Folin–Ciocâlteu reagent. Then approximately 300 µL Na_2_CO_3_ was added to the solution, which was then kept at 20 °C for 2 h. The absorbance was detected at 765 nm in comparison with a blank solution. A calibration curve was performed using different concentrations of gallic acid (*R*^2^ = 0.9672), and the results were expressed as mg gallic acid equivalents gallic acid GAE/g sample (DM).

### 2.13. Determination of Total Flavonoid Content

The total flavonoid content (TFC) of the tomato samples was determined following the method of Kim, Jeong, and Lee [30]. A mixture of methanolic extract (1 mL), 5% NaNO_2_ solution (300 μL), and 10% aluminum chloride (300 μL) was incubated at 25 °C for 5 min, and then 1 M sodium hydroxide (2 mL) was added. The volume was then directly adjusted to 10 mL with H_2_O and thoroughly vortexed. The absorbance was detected at 510 nm. A calibration curve was performed using different concentrations of catechin (*R*^2^ = 0.974). TFC was stated as mg catechin equivalents (CE)/g sample (DW).

### 2.14. 2,2-Diphenyl-1-Picrylhydrazyl Radical Scavenging

The radical scavenging (DPPH) radical scavenging ability of the fresh and solar dried tomato samples was determined as previously described [31]. Approximately 0.9 mL of a 50 mM Tris-HCl buffer (pH 7.4) and 0.1 mL of the sample extracts, or deionized H_2_O, to serve as a control, were mixed and then incubated at room temperature for 30 min. The absorbance of the mixture was then read at 517 nm. The DPPH scavenging was calculated and expressed as trolox equivalent (mg TE/g).

### 2.15. Statistical Analysis

Data are presented as arithmetic means of three replicates. The experiments were conducted using a completely randomized block design and analyzed using one-way analysis of variance. Multiple significant differences in the means (*p* < 0.05) were evaluated using least significant difference (LSD) range test. Multivariate analysis was conducted using HJ-Biplot principle component analysis (PCA) algorisms as described in the XSTAT software.

## 3. Results and Discussion

### 3.1. Moisture Content and Water Activity of Fresh and Solar-Dried Tomato Slices

The changes in the moisture content (MC), water activity (a_w_), and the total acidity of the solar-dried tomato and dried tomato followed by storage are shown in Table 1. The MC for the n fresh tomato MC was found to be 94.7%, and was sharply decreased to to 3.95% after solar drying, and remained at the same level after stored for 90 and 180 days (3.51%–3.44%). The insignificant change in the moisture content of dried-tomato slices during storage might be attributed to the vacuum packaging. Similarly, the fresh tomato showed significantly higher water activity (0.91) than solar-dried tomato before and after storage (0.31–0.42).

The low misture conent and the water activity in solar-dried samples could be attributed to the efficient and quick removal of water from tomato slices, due to the uniform heat distribution and transfer in the dryer. Similar findings were reported by Mdziniso et al. [32] in green leafy and yellow succulent vegetables upon drying and subsequent ambient storage. Indicating that the solar drying method is promising cheap technques for drying vegetables that is associated with lower MC and water activity.

### 3.2. Microbial Load of Fresh and Solar-Dried Tomato Slices

The results showed that the log values of *S. aureus* and *S. saprophyticus* were higher in fresh tomato compared with those in solar-dried tomato before and after the storage (Table 2) at 3.38 and 3.60 log cfu/g, respectively. However, after solar drying, the log values of both *S. aureus* and *S. saprophyticus* was reduced to 3.0 log cfu/g. In the stored solar dried tomato slices, the log values of the *S. aureus* reduced to 1.5 log cfu/g, whereas *S. saprophyticus* was totally eliminated. *E. coli* was not detected in either fresh or dried samples. However, yeast was detected in solar-dried tomato (5.09 log cfu/g) but not in fresh samples. This contamination of tomato slices might be occurred during the drying process.

Overall, the average microbial population was significantly (*p* < 0.05) lower in the solar-dried tomato slices, particularly after storage, compared to the fresh samples. Ntuli et al. [33], reported that the absence of microbes in detection procedures indicates good hygienic, drying, storage, and handling practices. The elimination or the lowering the load of *S. aureus*, *S. saprophyticus*, and *E. coli*, the mold and yeast in the solar-dried sample could be attributed to decreasing both the moisture content and the tomato samples water activity after drying. Moreover, the microorganisms can keep their viability regardless of the water activity. The growth of bacteria require a_w_ > 0.8 whiles yeasts and molds grow in a_w_ > 0.6 [33]. During the storage period, there was a very slight decrease in the microbial load in dried samples, indicating that microbial safty of the solar drying of tomato slices up to 180 days of storage period.

### 3.3. Color Values of Fresh and Solar-Dried Tomato Slices

Color is an important characteristic in determining the quality of various foods. Pigments and aroma may be formed through non-enzymatic browning reactions when food is dried. Table 3 shows the changes in color values of the fresh tomato and solar-dried tomato followed by storing. The color values *L*, *a*, and *b* of fresh tomato were 56.4, 18.4, and 39.4, respectively. The lightness (*L* value) of the tomato was significantly (*p* < 0.05) decreased to 30.0 after solar drying and progressively decreased to 27.4 and 24.9 when the solar-dried tomato was stored for 90 and 180 days, respectively. The redness (*a* values) of the tomato sample was significantly (*p* < 0.05) altered after solar drying to 15.6 (Table 3). The *a* value was also found influenced by storing time. The lowest value, 8.81, was found in solar-dried tomato stored for 180 days. Similarly, there was a significant (*p* < 0.05) decrease in the yellowness (*b* values) in the solar-dried samples alone and dried and storage samples. Among the dried samples, the lowest *b* value (11.4) was obtained after 180 days of storage.

We calculated the color difference Δ*E* to evaluate the change in color in the solar-dried tomato and solar-dried samples, followed by storage time. The Δ*E* was significantly (*p* < 0.05) higher in stored tomato compared to the samples that were not stored and were found to be 32.2, 39.7, and 43.3 in dried samples stored for 0, 90, and 180 days, respectively (Table 3).

Moreover, the solar drying and/or storage process caused a significant decrease in lightness (*L**), redness (*a**), and yellowness (*b**), leading to decreased in the BI for dried tomato. The BI was found to be 135.9 berior to the solar drying and it was decreased to 113.5 after drying of tomato slices. Additionally, storing of dried tomato slices for 90 and 180 day progressively decreased the BI to 94.7 and 85.3, respectively. The change in colour and BI of the dried sample was mainly due to the non-enzymatic browning/Maillard reaction. The decomposition of colour pigments and non-enzymatic reactions may occur and produce darkness in dried tomato slices [34]. Likewise, Ahmed et al. [35] and Chong et al. [36] reported the decrease in the *L** value with increase in the darkness/brownness of food materials and pigment destruction were stated by various researchers.

### 3.4. Total Acidity, Vitamin C, Total Carotenoid, and Lycopene Content of Fresh and Solar-Dried Tomato Slices

Table 4 shows the effect of solar drying alone or followed by storage on the total acidity, vitamin C content, total carotenoid, and lycopene content of fresh tomato. The fresh tomato’s total acidity was found to be 4.19%. Solar drying processing caused insignificant (*p* > 0.05) increased in the total acidity. However storing the solar treated tomato slices for 90 and 180 days significant (*p* < 0.05) increase total acidity to 8.83 and 8.87, respectively. Similarly, Abrol et al. [20] reported an increase in acidity after solar drying of mango, banana, and papaya fruits. The increase in acidity may be due to MC reduction, leading to concentration-effect that increased acidity. Furthermore, the increase in acidity could be attributed to acid formation fromsugars oxidation and other chemical reactions [20].

The solar drying significantly (*p* < 0.05) decreased the vitamin C content of the fresh tomato from 63.8 mg/g DM to 53.1 mg/g DM, 52.2, and to 52.1 mg/g DM for 0, 90, and 180 days, respectively. However, there were no difference (*p* < 0.05) in the vitamin C content of samples with the storage time of 90–180 days (Table 4). Abrol et al. [20] reported similar trend in vitamin C losses after solar drying of mango, banana, and papaya that reached 78%, 75%, and 77.9%, respectively. Moreover, Nyangena et al. [37] found that vitamin C content in mango slices decreased with increased drying temperature, with the highest value observed in fresh samples. Vitamin C is susceptible to heat, which may explain the reduction in content following solar drying. Studies have shown that retaining vitamin C is difficult due to its heat liability [20,38].

The total carotenoid content of fresh tomato slices was 41.9 mg/g DM. As illustrated in Table 4, the solar drying process caused a significant (*p* < 0.05) increase in the total level of carotenoids of the tomato slices to 63.7 mg/g DM. Interestingly, the total carotenoid content was significantly higher in stored dried tomato at 88.6 and 88.9 mg/g DM after 90 and 180 days storage, respectively. Although, thermal processing such as dehydration, cooking, and baking is known to reduce carotenoid levels [39]. However, solar drying in this study significantly (*p* < 0.05) increased the total carotenoid level in tomatoes, which may be due to the low temperature and short time applied for the drying process. Similar to our findings drying of tomato using different drying methods such as sun, hot air oven, and microwave oven drying caused a significant increasing on its total carotenoid levels [40,41].

Lycopene content of the tomato prior to the solar drying was found to be 26.8 mg/100 g DM (Table 4). The solar drying process significantly (*p* < 0.05) increased the lycopene content in tomato to 30.5 mg/100 g DM. Similar results were also stated by Xianquan et al. [38], who found that lycopene remains relatively stable during thermal food processing procedures, particularly at low temperature and short application time. The tomato’s thermal processing may cause an increase in the lycopene content in tomato after drying, improving the bioavailability of lycopene, the principal carotene in the fruit, by cell wall disruption [38]. However, the lycopene content in tomato slices decreased significantly (*p* < 0.05) in storage after 180 days (29.6 mg/100 g DM). Reduced lycopene content after a long time of storage, 180 days in this study, might be due to autoxidation, which may cause fragmentation of the lycopene molecules [42].

### 3.5. Total Phenolic and Flavonoid Content and Antioxidant Activity of Fresh and Solar-Dried Tomato Slices

The TPC, TFC, and antioxidant activity (AA) as DPPH scavenging activity of fresh and dried tomato are illustrated in Table 5. The TPC, TFC, and AA of fresh tomato were 2.11 mg GAE/g, 1.90 mg Que/g, and 2.71 mg trolox/g in DM, respectively. Solar drying (0 day) significantly incread TFA content but not TPC and AA. However storing solar dried tomato slices significantly increased the content of TPC and TFC and AA activity levelcompared to fresh tomato.There was laso significant differeces in TPC, TFC, and AA between tomato slices stored for 90 and 180 days. The highest values of the TPC (3.18 mg GAE/g), TFC (4.48 mg Que/g), and AA (3.55 mg trolox/g) were observed after 180 days of storage. Similarly, several studies have reported that the solar drying process significantly increased the content of phenolic compounds and AA in some fruit and vegetables [20,21]. This increase was principally due to the increased release of phytochemicals from the food matrix, which increased these components’ extraction [43]. Moreover, the enhancement of phenolic compound content and the AA of solar-dried tomato may be attributed to the release of bound phenolic compounds resulting from cellular constituents’ breakdown due to the fast drying process [44,45].

Thermal treatment, such as oven drying, microwave heating, and sun drying, sharply reduced phenolic compounds and AA in dried leaf vegetables [31]. However, this study revealed that drying tomato slices using solar energy dryers enhanced the phenolic and flavonoid content and the AA. So, the increasing on the antioxidant activity might also be due to the hydrogen donation ability of antioxidants occurred as influenced by solar drying [46].

### 3.6. Principal Component Analysis and Partial Least Squares Regression Analysis

PCA using the HJ-Biplot method was performed to show the interrelationships between solar drying, storage, and measured parameters. The results demonstrated the interactive effects of the solar drying process on the physicochemical characteristics and phenolic compound contents and the microbial load and AA of solar-dried tomato, particularly after storage for 90 and 180 days (Figure 1). Interestingly, the contribution of the axes of the principal components PC1 and PC2 is explained as 79.4% and 17.6%, respectively, which resulted in high variability (97.0%) of the plotted components. Moreover, there was a strong positive correlation between radiation treatments and quality parameters of spices. Since s is an acute angle was found between the vectors of these parameters. Yan and Fregeau–Reid [47] explained that the eccentricity of characters that appear <90° angle is positively correlated, whereas those factors that formed >90° angles were associated with negative correlation, and those who have a 90° angle do not show a correlation in the biplot. Consequently, the treatments were separated into three groups. The control sample (fresh fruit) had higher values for vitamin C content and water activity of the tomato slices. The solar-dried tomato fruit (SDF) stored for 90 and 180 days (SDF 90 days and SDF 180 days) had the highest degree of correlation between the treatment parameters (lycopene content, color, carotenoids, TPC, TFC, acidity, and AA). Accordingly, these findings revealed that storing solar-dried tomato slices up to 6 months at 25 °C improve storability, physicochemical properties, and antioxidant capacity, hence extending shelf life.

The partial least squares regression (PLS) analysis classified the solar dying process’s validation on the storability and antioxidant characteristics of tomato slices. The interactive effects of solar drying and the storage time (*x*-variables) on the fungal load, color, water activity, total acidity, vitamin C, total carotenoid, lycopene content, TPC, TFC, and AA (*y*-variables) of tomato slices were clearly observed (Figure 2). The PLS revealed that solar-dried tomato slices particularly those stored for 90 and 180 days, showed positive validation of most studied parameters. Hence, the PLS specified that solar energy application in tomato drying followed by storage at ambient conditions produced the most valid drying treatment that could be considered for food industry applications.

## 4. Conclusions

This study investigated changes to the microbial load, physicochemical characteristics, and antioxidant capacity of solar-dried tomato slices during storage. The findings revealed that solar drying enhanced the content of bioactive compounds and the tomato slices antioxidant capacity and can be a beneficial factor for reducing the *S. aureus*, *S. saprophyticus*, and *E. coli*, the mold and yeast load of dried tomatoes and enhancing storage properties. Thus, low-tech solar energy shows great potential for food industry applications. It could become an emerging effective post-harvest preservative method. However, further research on application of solar drying technology concerning the impact on other quality parameters, and the storage properties of other fruits and vegetables are needed to improve its efficiency to accelerate the drying processing as well as to inhibit the microbial contamination during drying before it can be used as a reasonable low-tech alternative effective preservative method in food industries and technology.

## Figures and Tables

**Figure 1 foods-10-00734-f001:**
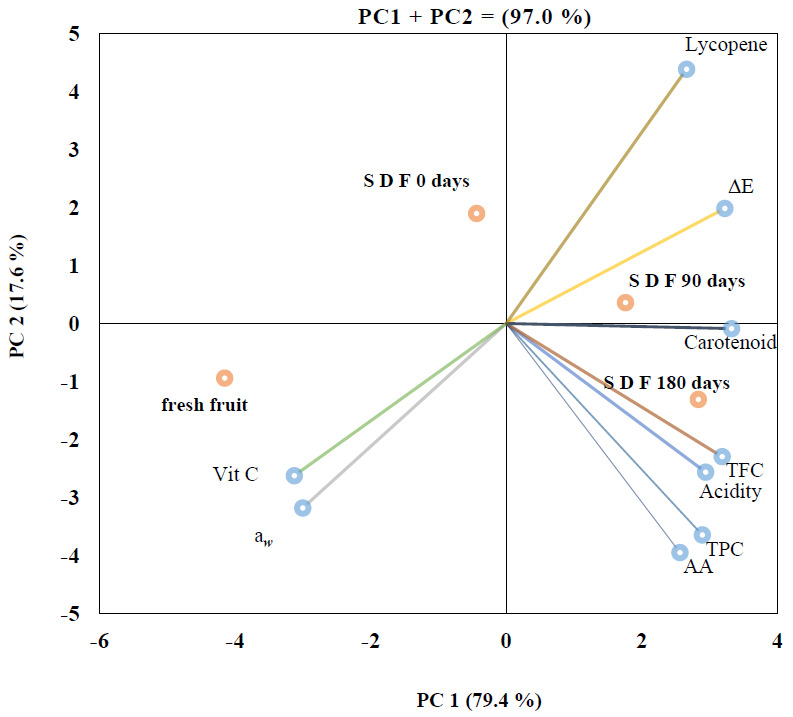
Principal component analysis (PCA) of color (∆E), a_w_, total acidity, vitamin C content, total carotenoids, lycopene content, TPC, TFC, and AA of fresh fruit, solar-dried fruit (S D F 0 days), solar-dried fruit stored for 90 days (S D F 90 days), and 180 days (S D F 180 days).

**Figure 2 foods-10-00734-f002:**
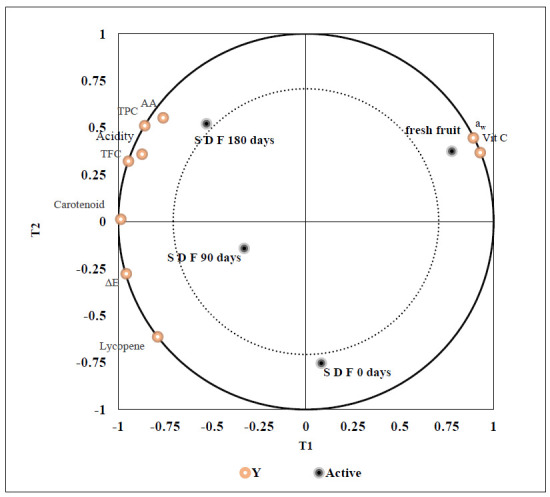
Partial least squares regression analysis (PLS) of color (∆E), a_w_, total acidity, vitamin C content, total carotenoids, lycopene content, TPC, TFC, and AA of fresh fruit, solar-dried fruit (S D F 0 days), solar-dried fruit stored for 90 days (S D F 90 days), and 180 days (S D F 180 days).

**Table 1 foods-10-00734-t001:** Moisture content (%) and water activity (a_w_) of fresh and solar-dried tomato slices.

Parameters	Fresh Fruit	Stored Solar-Dried Fruit		
		0 Day	90 Days	180 Days
MC	94.42 ± 0.28 ^a^	3.93 ± 0.57 ^b^	3.51 ± 0.30 ^b^	3.44 ± 0.33 ^b^
a_w_	0.93 ± 0.02 ^a^	0.31 ± 0.001 ^b^	0.31 ± 0.002 ^b^	0.32 ± 0.001 ^b^

Means in the same raw with the same superscript are not significantly (*p* < 0.05) different as assessed by least significant difference (LSD).

**Table 2 foods-10-00734-t002:** Microbial load of fresh and solar-dried tomato slices.

Microorganism Log cfu/g	Fresh Fruit	Stored Solar-Dried Fruit		
		0 Day	90 Days	180 Days
*Staphylococcus aureus*	3.38 ± 0.40 ^a^	3.0 ± 0.01 ^b^	1.5 ± 0.03 ^c^	1.5 ± 0.01 ^c^
*Staphylococcus saprophyticus*	3.60 ± 0.02 ^a^	3.0 ± 0.01 ^b^	0.0 ± 0.00 ^c^	0.0 ± 0.00 ^c^
*Escherichia coli*	nd	nd	nd	nd
Mold	nd	nd	nd	nd
Yeast	nd	5.09 ± 0.09 ^a^	nd	nd

Means in the same raw with the same superscript are not significantly (*p* < 0.05) different as assessed by LSD. nd: Not detected.

**Table 3 foods-10-00734-t003:** Color values of fresh and solar-dried tomato slices.

Parameters	Fresh Fruit	Stored Solar-Dried Fruit		
		0 Day	90 Days	180 Days
*L**	56.4 ± 0.84 ^a^	30.0 ± 0.73 ^b^	27.4 ± 1.31 ^c^	24.9 ± 0.90 ^d^
*a**	18.4 ± 0.22 ^a^	15.6 ± 0.73 ^b^	11.7 ± 1.00 ^c^	8.81 ± 1.15 ^d^
*b**	39.4 ± 0.02 ^a^	16.4 ± 0.76 ^b^	13.2 ± 1.60 ^c^	11.4 ± 0.50 ^d^
Δ*E*	0.00 ± 0.00 ^d^	35.2 ± 0.74 ^c^	39.7 ± 1.30 ^b^	43.3 ± 0.85 ^a^
BI	135.9 ± 0.00 ^a^	113.5 ± 0.09 ^b^	94.7 ± 0.97 ^c^	85.3 ± 0.89 ^d^

Means in the same raw with the same superscript are not significantly (*p* < 0.05) different as assessed by LSD.

**Table 4 foods-10-00734-t004:** Total acidity (%, DM), vitamin C (mg/g, DM), total carotenoid (mg/g, DM), and lycopene content (mg/100 g, DM) of fresh and solar-dried tomato slices.

Parameters	Fresh Fruit	Stored Solar-Dried Fruit		
		0 Day	90 Days	180 Days
Total acidity	4.19 ± 0.19 ^b^	4.37 ± 0.05 ^b^	8.83 ± 0.29 ^a^	8.87 ± 0.23 ^a^
Vitamin C	63.6 ± 0.126 ^a^	53.1 ± 0.46 ^b^	52.2 ± 0.53 ^c^	52.1 ± 0.16 ^c^
Carotenoid	41.9 ± 0.42 ^c^	63.7 ± 0.15 ^b^	88.6 ± 0.08 ^a^	88.9 ± 0.10 ^a^
Lycopene	26.8 ± 0.45 ^c^	30.5 ± 0.04 ^a^	30.4 ± 0.08 ^a^	29.6 ± 0.06 ^b^

Means in the same raw with the same superscript are not significantly (*p* < 0.05) different as assessed by LSD.

**Table 5 foods-10-00734-t005:** Total phenolic content (TPC; mg GAE/g DM) and total flavonoid content (TFC; mg Que/g DM) content and antioxidant activity (AA; 2,2-diphenyl-1-picrylhydrazyl (DPPH) mg trolox/g DM) of fresh and solar-dried tomato slices.

Parameters	Fresh Fruit	Stored Solar-Dried Fruit		
		0 Day	90 Days	180 Days
TPC	2.11 ± 0.12 ^c^	2.15 ± 0.07 ^c^	2.68 ± 0.20 ^b^	3.18 ± 0.27 ^a^
TFC	1.90 ± 0.26 ^d^	2.46 ± 0.09 ^c^	3.77 ± 0.23 ^b^	4.48 ± 0.53 ^a^
AA	2.71 ± 0.03 ^c^	2. 75 ± 0.06 ^c^	2.91 ± 0.00 ^b^	3.55 ± 0.10 ^a^

Means in the same raw with the same superscript are not significantly (*p* < 0.05) different as assessed by LSD.

## Data Availability

Data is contained within the article.

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
