# Peer review of "The Potential of Exploiting Economical Solar Dryer in Food Preservation: Storability, Physicochemical Properties, and Antioxidant Capacity of Solar-Dried Tomato (Solanum lycopersicum) Fruits"

_foods, 2021, doi:10.3390/foods10040734_

Round 1
Reviewer 1 Report
The authors report a work about the potential of using solar energy to dry tomatoes. The work is interesting and it poses consideration about the use of such technology especially in emerging countries. In my opinion, the author should have included control samples to compare the technology with a gold standard. If not applied a better comparison and discussion with literature should be included for all the chemicals.
In the instruction a better focus on innovative drying technologies with their limitation should be included.
What is the current shelf-life of dried tomatoes? Why did you finish your investigation at 6 month? Please include this information in discussion/conclusion with references.
In the supplementary picture of the product should be included at different time point to support data on colour.
About the microbial inactivation, it is known the resistance of some pathogens in the dried state. Based on your methodology and the results you can not say that the treatment eliminated microorganisms. The authors only confirmed the current presence of microorganism in the product before and after the drying. In the case of yeasts and molds they were found only in the dried products and not in the fresh one. For this reason, I think results are effected by uncertainty due to the initial natural contamination. Challenge test should be performed to demonstrate the actual inactivation of specific microorganisms. I suggest to modify the abstract and the results or include challenge tests.
Methods:
The growing temperature should not be the same for all the microorganisms. Colony should be counted after fixed days of incubation. Moreover the CFU should be normalized by the initial weight of the sample. Please clarify the methods.
Line 217. Did you perform storage at 10 and 30 days or 90 and 180 days?
Author Response
Responses to the reviewer’s comments
Reviewer 1
Dear Sir:
Thank you very much for your evaluation and valuable comments on our manuscript (foods-1124079). In response to your kind and helpful suggestions, we have revised our manuscript and the changes we made are highlighted in red in the manuscript text. Your comments, the revisions made, and our responses to your remarks are listed in the attached file
regards

Reviewer 2 Report
The work is interesting and has a practical value, but it requires clarification, especially with regard to drying conditions. Below are my questions to the authors.
Where and when were the tomatoes washed? After harvest or just before drying? How long did it take from harvesting to drying? Where did the drying of the tomatoes take place (in Sudan?) and how far from the place of their harvest? What was the section plane (cut of the tomato) - longitudinal or wide? Were the dried tomatoes vacuum packed into bags? If not, could the dried product come into contact with oxygen during storage (contained inside the bags during packaging or due to possible leakage in the bags)?
Which light source of colorimeter was used?
What do the numbers 90 and 91 on lines 116 and 117 mean?
Please indicate the literature source or indicate as your own method for the equation for "browning index".
The solar drying process is incomplete. There is no information about the dimensions of the tunnel, whether there was tunnel ventilation, if so, what was the air exchange per unit of time, what was the transparency of the material for sunlight, whether the material from which the tunnel was made was permeable or impermeable to UV radiation, or fruit slices were rotated whether or not the maintenance of the indicated temperature was supported at night, etc. Perhaps placing a photo of the drying process in the tunnel would be beneficial for understanding the specifics and realities of this process.
In tables 2, 3, 4 and 5 it should be "Means in the same rows", not "Means in the same column".
Author Response
Responses to the reviewer’s comments
Reviewer 2
Dear Sir:
Thank you very much for your evaluation and valuable comments on our manuscript (foods-1124079). In response to your kind and helpful suggestions, we have revised our manuscript and the changes we made are highlighted in red in the manuscript text. Your comments, the revisions made, and our responses to your remarks are listed in the attached file
regards

Reviewer 3 Report
The reviewed paper entitled “The potential of exploiting low-tech solar energy in food preservation: Storability, physicochemical properties, and antioxidant capacity of solar-dried tomatoes(Solanum Lycopersicum) fruit” presents a somewhat original research with interesting results. However, there are some issue deserving modifications, clarifications, and reconsiderations. Therefore, major revision instead of present form needs to be made prior to consideration for being published in Foods MDPI.
The reviewer’s comments are listed as followings:
Title:
The authors need to consider to address “low-tech” alternatively; “tech” is not generally appeared in an academic article; may it be “basic”, “simplified“, “economical type” or whichever suited better. The scientific name “lycopersicum” is not to be capitalized.
Abstract:
The results regarding the PCA analyses are interesting however and deserve to be included in the section of abstract.
Introduction:
I. There is a need to literally differentiate the solar and sun drying in the paragraph started form line 52
II. The justified goal of the present research started from line 68 are not solid enough. There had been published articles reporting the solar drying application with regards to tomato preservation; some amongst publications even revealed the antioxidant content variations due solar drying. Therefore, the novelty of the article become questionable. Readers would possibly become negligible to the authors’ motivation to systematically study parameter correlation with PCA which features a dimension reduction approach. Please elaborate.
Materials and methods
Some information in section 2.3 is to be supplemented:
I. The cross section area/major contact surface in terms of unit-length-square.
A. The sample placement and arrangement,
B. The air velocity or the heat transfer coefficient inside the tunnel unit,
C. A schematic of the solar drying unit should be supplemented in this section, and,
D. Some pictures of of sliced tomato samples arranging on the drying tray and conveyer belt assembling as one figure would be of great help for the reader to visualize the the drying process.
II. Please avoid alternating usage of “mold” and “fungal/fungus” in section 2.4 and throughout the entire manuscript.
III. In section 2.5, the manufacturer of the water activity apparatus should be provided. The corresponding temperature upon water activity should also be
mentioned. Other than the measurement of the moisture content, is it possible to a mathematical model to describe the mass transfer behaviors or dry kinetics of the solar drying process?
IV. There are lots of errors in section 2.7: The star sign, the numbers (90, 91, and 92), the conversion 2ΔL/Δa/Δb, and the reference of BI.
V. The manufacturer of the absorbance detector should be provided.
VI. In the subheading of 2.11, it is suggested that the DPPH antioxidant activity is to be directly stated. Also in the section text, the full chemical description of DPPH should be addressed.
VII. In section 2.15, please be specific regarding the significant difference ranges test, i.e. Duncan’s/post hoc/turkey’s.
Results and discussion
General:
I. Please arrange the results presentation consistent/parallel to Materials and Methods, for example “moisture content and water activity measurements” should be prior to “microbial load”
II. In section 3.1, 3.2 and 3.3, the results should be well discussed; the authors should carefully explain their results and compare with existing studies.
III. The statistical significance states in the table footnotes are confusing or even incorrect; there are no superscript letters presented in any of the tables; the comparisons are to be associated with the means in rows.
Specific:
I. In section 3.1, the unusual microbial load of Yeast counts for the dried sample on day zero would need to be justified for possible causes.
II. In section 3.2 Line 217, “…10 and 30 days…” must be a made error, should be 90 and 180 days. Other than the numerical error, the moisture content literally deceased during the storage there should some insight discussion to elaborate.
III. In section 3.3 Line 248, again, the incorrect measurement days are reported. Furthermore, the values of BI are only reported without further discussion.
IV. In section 3.4, please carefully illustrate the mechanism of drying effect upon promoting the contents of carotenoid and lycopene; there should be sufficient reported available. Crucially, the last sentence of this section between Line 285 and 286 should be edited in the light of clarity: increment?, concerning?
V. In section 3.5, the authors consider that the increases of antioxidant related components and activity is based the release-extraction causation; please elaborate some more possible aspects. The provided discussion along with leafy vegetables could be a proper analogue, however, some additional writeup should be supplemented.
VI. In section 3.6, for the PCA analysis, it is necessary to scrutinize the possible attributes of PC1 and PC2. Does authors have any clue regarding this issue? Some discussion to the dimension reduction statistical model would make this present article much worth reading. Is it possible to inclusion the clustered data range as shown in literature such as https://doi.org/10.1016/j.ifset.2020.102434. The reviewer is only familiar with a free software, namely, R statistics. The commends of “ggplot 2” library is available for the data clustering. Please do the same presentation using XSTAT if possible. Additionally, there is a noted blue circle at the bottom Figure 2, however, not shown the corresponding data in the figure itself; please verify!
References:
Listed references are to be numbered as per the designated format guideline.
The English language:
General:
The authors have supplemented a certificate of editing for English language, grammar, punctuation, and spelling. The reviewer also recognized the endeavors that the agency had provided. However, the written content in its current form is evidence of many logical fallacies to the potential readers who are food science and technology professional. The followings are some suggestions but not limited to the rest of the part article for a better proof of professional understandings.
I. The authors have frequently used “apostrophe s” to indicate the ownership or show possession of object nouns. Please use apostrophe s less frequently in the text editing.
II. “Tomatoes fruit” is to be rephrased as “sliced tomatoes” or ”tomato slices”.
III. The statistical significance of p should be always in small italic through the text.
Some specifics but not limited:
Line 29, capitalization to edit.
Line 29-30, “…as essential nutritional and health sources…” edition needed.
Line 31-32, the listed nutrients and compounds are to be organized systemically.
Line 32-33, “A correlation …”, try to avoid the nesting structure of “and”.
Line 41, so as to extend...
Line 44, “…are lost.” edition needed
Line 44-45 ? …suitable preservation approaches with low-cost processing features should…
Line 50-51 “Furthermore, …” edition needed.
Author Response
Responses to the reviewer’s comments
Reviewer 3
Dear Sir:
Thank you very much for your evaluation and valuable comments on our manuscript (foods-1124079). In response to your kind and helpful suggestions, we have revised our manuscript and the changes we made are highlighted in red in the manuscript text. Your comments, the revisions made, and our responses to your remarks are listed in the attached file
regards

Round 2
Reviewer 1 Report
The authors answered sufficiently to my previous comments.
Author Response
Responses to the reviewer’s comments
Reviewer 1
Dear Sir:
Thank you very much for your evaluation and valuable comments on our manuscript (foods-1124079R1). In response to your kind and helpful suggestions, the details of the drying process and storage of solar-dried tomato were added to text as you requested. Also the methods was written in details.
Thank you very much
Sincerely Yours
Dr. Amro B. Hassan
The Corresponding author
Reviewer 3 Report
The quality of the revised manuscript has significantly improved. The review read an interest and comprehensive work. Please conduct the minor editions throughout the text with respect to the unit expression and the city location of the apparatus manufacturers. Some specific text that needs to be edited are listed but not limited:
Line 94 the unit is to be super-scripted.
Line 128 "92" is to be removed. The apparatus information would possibly be “a colormeter (CR-300, Konica Minolta Inc., Tokyo, Japan)”.
The expression of the temperatures with variations throughout the text are not consistent regarding the spaces in between; they are to be unified.
Author Response
Responses to the reviewer’s comments
Reviewer 3
Dear Sir:
Thank you very much for your evaluation and valuable comments on our manuscript (foods-1124079R1). In response to your kind and helpful suggestions, we have revised our manuscript and the changes we made are highlighted in red in the manuscript text. Your comments, the revisions made, and our responses to your remarks are listed below:
Line 94 the unit is to be super-scripted.
Response: Done
Line 128 "92" is to be removed. The apparatus information would possibly be “a colormeter (CR-300, Konica Minolta Inc., Tokyo, Japan)”.
Response: Corrected
The expression of the temperatures with variations throughout the text are not consistent regarding the spaces in between; they are to be unified.
Response: Done
Thank you very much
Sincerely Yours
Dr. Amro B. Hassan
The Corresponding author
